# QUANTUM ARCHITECTURE SEARCH WITH UNSUPERVISED REPRESENTATION LEARNING

## ABSTRACT

Utilizing unsupervised representation learning for *quantum architecture search* (QAS) represents a cutting-edge approach poised to realize potential quantum advantage on *Noisy Intermediate-Scale Quantum* (NISQ) devices. QAS is a scheme to design quantum circuits for *variational quantum algorithms* (VQAs). Most QAS algorithms combine their search space and search algorithms together and thus generally require evaluating a large number of quantum circuits during the search process, which results in formidable computational demands and limits their applications to large-scale quantum circuits. Predictor-based QAS algorithms can alleviate this problem by directly estimating the performance of circuits according to their structures. However, a high-performance predictor generally requires very time-consuming labeling to obtain a large number of labeled quantum circuits because the gate parameters of quantum circuits need to be optimized until convergence to their ground-truth performances. Recently, a classical neural architecture search algorithm *Arch2vec* inspires us by showing that architecture search can benefit from decoupling unsupervised representation learning from the search process. Whether unsupervised representation learning can help QAS without any predictor is still an open topic. In this work, we propose a framework QAS with unsupervised representation learning and visualize how unsupervised architecture representation learning encourages quantum circuit architectures with similar connections and operators to cluster together. Specifically, our framework enables the process of QAS to be decoupled from unsupervised architecture representation learning so that the learned representation can be directly applied to different downstream applications. Furthermore, our framework is predictor-free eliminating the need for a large number of labeled quantum circuits. During the search process, we use two algorithms *REINFORCE* and *Bayesian Optimization* to directly search on the latent representation, and compare them with the method *Random Search*. The results show our framework can more efficiently get well-performing candidate circuits within a limited number of searches.

## 1 INTRODUCTION

Quantum Computing (QC) has made progress in the last decades. The development of quantum hardware and new quantum algorithms have shown their potential to provide advantages over classical computers in diverse tasks, such as image processing (Wang et al., 2022), reinforcement learning (Skolik et al., 2022), knowledge graph embedding (Ma et al., 2019), and network architecture search (Zhang et al., 2022; Giovagnoli et al., 2023; Du et al., 2022). However, the scale of a quantum computer is still limited by the noise, which comes from the environment and leads to an unstable performance. These noisy intermediate-scale quantum (NISQ) devices fall short of fault tolerance in the near future (Preskill, 2018). The variational quantum algorithm (VQA), a hybrid quantum algorithm using quantum operations with adjustable parameters, is referred to as a leading strategy in the NISQ era (Cerezo et al., 2021). In VQA, the parameterized quantum circuit (PQC) with trainable parameters is viewed as a general paradigm of quantum neural networks and has reached remarkable achievements in quantum machine learning. These parameters control quantum circuit operations, changing the distribution of circuit output states, and are updated by a classical optimizer given a task-specific objective function. Although VQA has its own problems such as Barren Plateaus (BP) or scalable problems, it has shown the potential to improve performance by solving diverse problems

such as image processing, combinatorial optimization problems, chemistry, and physics (Pramanik et al., 2022; Amaro et al., 2022; Tilly et al., 2022). Variational quantum eigensolver (VQE) (Peruzzo et al., 2014; Tilly et al., 2022) is one of VQAs. It is referred to as an approximator of the ground state and provides more flexibility for quantum machine learning. We are considering using VQE for some circuit performance evaluation.

Unsupervised representation learning seeks to find hidden patterns or structures of unlabeled data, and it is a well-studied problem in general computer vision research (Radford et al., 2015). Autoencoder as one of the approaches for unsupervised representation learning has good expressiveness for feature representation. It consists of an encoder and decoder, which first maps images into a compact feature space and then decodes features to reconstruct similar images. Besides images, the autoencoder can learn good features from graphs, such as encoding and reconstructing a directed acyclic graph (DAG) or a neural network architecture (Yan et al., 2020; Zhang et al., 2019; Pan et al., 2018; Wang et al., 2016). Architecture search and representation learning are coupled in most research. However, this leads to an inefficient search, which strongly depends on labeled architectures by numerous evaluations. A framework *Arch2vec* seeks to make the representation learning independent of architecture search by downstream conducting search algorithms (Yan et al., 2020). This decoupling helps to reach a smooth latent space that can benefit different search algorithms without numbers of labeling.

Quantum architecture search (QAS) or quantum circuit architecture search is a scheme to design quantum circuits efficiently and automatically. It aims to create a circuit to increase the learning performance (Du et al., 2022). Diverse algorithms are proposed for QAS (Zhang et al., 2022; Du et al., 2022; Zhang et al., 2021; He et al., 2023a; Giovagnoli et al., 2023). However, most algorithms define the search space and search algorithm together, leading to inefficient performance and high evaluation costs. The search algorithm performance often depends on how well the search space is defined, embedded, and learned. In order to find a suitable circuit, they need to evaluate different architectures many times. Although a predictor-based QAS He et al. (2023a) can separate the representation learning and search algorithm, it must often label different architectures by evaluation, and the training performance strongly depends on the quantity and quality of evaluations as well as the embedding. In this work, we are inspired by the idea of decoupling, and try to conduct QAS without any labeling. We seek to find out whether the decoupling can help to embed quantum circuit architectures into a smooth latent space benefiting predictor-free QAS algorithms.

We summarise our contributions as follows:

1. We conduct a general framework for QAS by decoupling the unsupervised architecture representation learning from QAS. Without labeling, this framework can embed quantum architectures into a compact, smooth latent space.

2. We use two popular visualization approaches PCA (Shlens, 2014) and t-SNE (Van der Maaten & Hinton, 2008) to demonstrate high-dimensional latent vector space and analyze the effect of their clustering. Both of them especially PCA can map high-performance quantum circuit architectures into a concentrated region.

3. After acquiring the latent representation, we apply REINFORCE and Bayesian optimization approaches directly to it during the search process. In this way, our framework eliminates the need for pre-training a high-performance predictor with a large number of labeled circuits, and it is free from the uncertainty resulting from prediction.

4. We conduct various experiments in the field of QC including unitary approximation, maxcut, and quantum chemistry (Liang et al., 2019; Poljak & Rendl, 1995; Tilly et al., 2022) to show the effectiveness of our framework and that the pre-trained quantum architecture embedding can benefit QAS for these applications.

## 2 RELATED WORK

**Unsupervised Graph Representation Learning.**   Graph data is rapidly becoming a key instrument for understanding complex interactions among real-world objects, for instance, biochemical molecules (Jiang et al., 2021), social networks (Shen et al., 2023), purchase networks from e-buy websites (Li et al., 2021), and academic collaboration networks (Newman, 2001). Graph data is often represented as a discrete data structure, making it extremely difficult to solve downstream

tasks with large search spaces. Our work is based on unsupervised graph representation learning that aims to learn a low-dimensional compact and continuous graph embedding without supervision while preserving graph topological structures and node attributive features. In this domain, Perozzi et al. (2014); Wang et al. (2016); Grover & Leskovec (2016); Tang et al. (2015) propose a type of method using local random walk statistics or matrix factorization-based learning objectives to learn corresponding representations, Kipf & Welling (2016); Hamilton et al. (2017) propose a kind of method reconstructing the adjacency matrix of graphs by predicting edge existence, and Veličković et al. (2018); Sun et al. (2019); Peng et al. (2020) propose another type of method maximizing the mutual information between local node representations and a pooled graph representation, etc. Additionally, Xu et al. (2019) explores the expressiveness of Graph Neural Networks (GNNs) in terms of their capability to distinguish any two graphs and introduces Graph Isomorphism Networks (GINs) which are proven to be as powerful as the Weisfeiler-Lehman test (Leman & Weisfeiler, 1968) for graph isomorphism. Getting inspired by the success of *Arch2vec* (Yan et al., 2020), a method with unsupervised graph representation learning and used in classic neural architecture search (NAS), we injectively encode quantum architecture structures using GINs as well since quantum circuit architectures also can be represented as DAGs.

**Quantum Architecture Search (QAS).** As mentioned in the previous section, PQCs are required as ansatz for various VQAs (Benedetti et al., 2019). Expressive power and entangling capacity of PQCs play an essential role in their optimization performance (Sim et al., 2019). It is highly possible that a badly designed ansatz suffers from limited expressive power or entangling capacity, which results in the global minimum for an optimization problem out of reach. Furthermore, such ansatz may be more susceptible to noises (Stilck França & Garcia-Patron, 2021), waste quantum resources, or lead to barren plateaus that frustrate the optimization procedure (McClean et al., 2018; Wang et al., 2021). Therefore, a systematic approach namely QAS is proposed to search for optimal PQCs. The goal of QAS is to automatically, effectively, and efficiently search high-performance custom quantum circuits for given problems that not only minimize the loss functions but also satisfy some other constraints imposed by the hardware connections between qubits, the native set of quantum gates, the quantum noise model, the training loss landscape and other practical issues. Quantum architectures and neural network architectures have many similar properties, like hierarchical, directed, and acyclic structures, so existing QAS works have been heavily inspired by ideas from NAS. More specifically, greedy approaches (Mitarai et al., 2018; Tang et al., 2021), evolutionary or genetic methodologies (Zhang & Zhao, 2022; Ding & Spector, 2022), reinforcement learning (RL) engine-based methods (Kuo et al., 2021; Ostaszewski et al., 2021), Bayesian optimization (Duong et al., 2022), and gradient-based approaches (Zhang et al., 2022) have all been adopted to discover better PQCs for VQAs, but they require the evaluation of numerous quantum circuits during the search process, which is time-consuming and computationally demanding. Therefore, predictor-based approaches (Zhang et al., 2021; He et al., 2023b) are proposed to alleviate this problem, but they still suffer the aforementioned problem since they require a large number of labeled circuits to train a predictor with generalized capability. Furthermore, the predictor leads to additional uncertainty in QAS, so it is necessary to reevaluate the circuit candidates obtained by the predictor. In this work, we propose a framework to further improve the problems.

## 3 QAS with Unsupervised Representation Learning

In this work, we show our method in Figure 1, which consists of two independent learning parts: one is an autoencoder for circuit architecture representation learning, and another is the part for the search process, including search and estimation strategy. The number of gates in a circuit and an operation pool define our search space created by gate types such as {X, Y, Z, H, Rx, Ry, Rz, U3, CNOT, CZ, SWAP}. A random circuit generator provides a set of circuit architectures with the predefined qubit quantity, gate quantity, and maximal circuit depth, then encodes these architectures into two matrices, and feeds them into an autoencoder. The autoencoder independently learns a latent distribution from the search space and provides pre-trained architecture embeddings for search algorithms. The estimation strategy takes circuit architecture from the search algorithm and returns evaluation performance. We use the ground state of Hamiltonian to evaluate a circuit architecture for solving max-cut and quantum chemistry problems and use fidelity for unitary approximation problems.

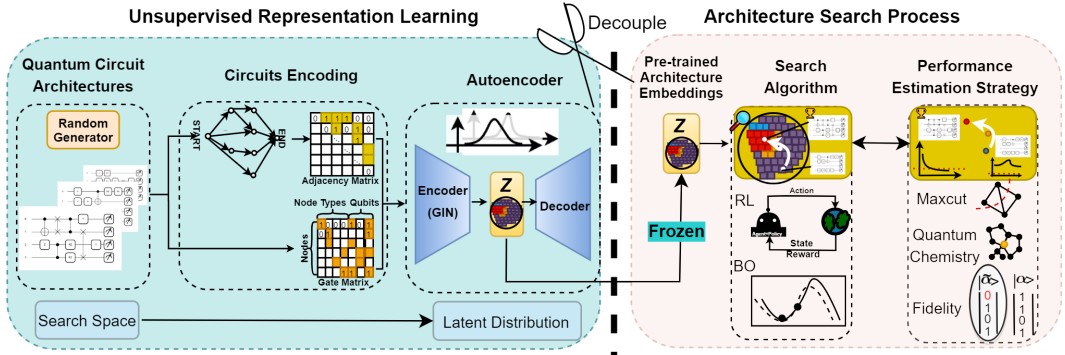

Figure 1: Illustration of the algorithm

## 3.1 CIRCUIT ENCODING SCHEME

We represent a quantum circuit with a DAG by a circuit encoding scheme described in He et al. (2023b;a). As known, any quantum circuit is a sequence of quantum gates. Each circuit can be transferred into a DAG by mapping gates on each qubit into a sequence of nodes, adding two nodes at the start and end of the sequence as circuit in-/output, and connecting nodes of each qubit from input to output according to their sequence order. Given a set of gates $O = \{gate_1, .., gate_k\}$, the mapping of a circuit of n qubits is shown in Figure 2a. An adjacency matrix will then describe the created DAG in Figure 2b. The set of nodes is additionally described through a gate matrix, which shows the node feature information. In Figure 2b, each row of the gate matrix stands for a node on the graph, the columns before the dotted line with one-hot encoding indicate the node type and other columns are used to encode position information such as which qubits the gate acts on. The encoding scheme still has some limitations whose details are introduced in Appendix A.1.

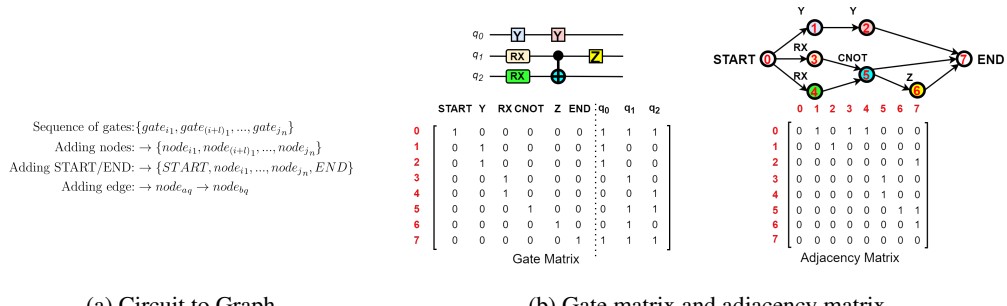

(a) Circuit to Graph

(b) Gate matrix and adjacency matrix

Figure 2: Circuit encoding scheme

## 3.2 VARIATIONAL GRAPH ISOMORPHISM AUTOENCODER

### 3.2.1 PRELIMINARIES

The most common graph autoencoders (GAEs) consist of an encoder and a decoder, which first maps a graph into a feature space and then decodes features to reconstruct a graph. As one of the autoencoders, the variational graph autoencoder (VGAE) is a promising framework for unsupervised graph representation learning, using a graph convolutional network encoder and a simple inner product decoder (Kipf & Welling, 2016). This work does not use VGAE as a framework to learn latent representations but with a powerful encoder, graph isomorphism network (GIN) (Xu et al., 2019).

**Definition 1.** *We are given a circuit created by $m$ gate types, $h$ gates and $g$ qubits. Then, the circuit can be described by a DAG $G = \{V, E\}$ with $n = h + 2 = |V|$ gate nodes including START and*

*END. The adjacency matrix of graph $G$ is summarized in $n \times n$ matrix $A$ and its gate matrix $X$ is in size of $n \times (m + 2 + g)$. We further introduce $d$-dimensional latent variables $z_i$ composing latent matrix $Z = \{z_1, .., z_K\}^T$.*

### 3.2.2 ENCODER

The encoder GIN maps the structure and node features to latent representations $Z$. An approximation of the posterior distribution $q(Z|X, A)$ is:

$$q(Z|X, A) = \prod_{i=1}^{K} q(z_i|X, A), \tag{1}$$

where $q(z_i|X, A) = \mathcal{N}(z_i|\mu_i, \text{diag}(\sigma_i^2))$. The $L$-layer GIN generates the embedding matrix $M^{(s)}$ for $s$-layer by:

$$M^{(s)} = MLP^{(s)}((1 + \epsilon^{(s)}) \cdot M^{(s-1)} + \hat{A}M^{(s-1)}), s = 1, 2, ..., L, \tag{2}$$

where $M^{(0)} = X$, and $\epsilon^{(s)}$ is a bias with the standard norm distribution for each layer. $MLP$ is a multi-layer perception consisting of Linear-Batchnorm-ReLU and $\hat{A} = A + A^T$ transforms a directed graph into an undirected one to capture bi-directional information. Then, the mean $\mu = \text{GIN}_\mu(X, \hat{A}) = FC(M^{(L)})$ is calculated by two fully connected layers $FC$, similarly for the standard deviation $\sigma$. We can sample the latent matrix $Z \sim q(Z|X, A)$ by $z_i = \mu_i + \sigma_i \cdot \epsilon_i$. For 4-qubit experiments, this work uses $L = 5$, a 16-dimensional latent vector $z_i$, and the GIN with hidden sizes of $\{128, 128, 128, 128, 16\}$ as the encoder and a one-layer MLP with a hidden dimension of 16 as the decoder. For 8-qubit experiments, we change the dimension of the latent vector to 32, the hidden sizes of the encoder to $\{128, 128, 128, 128, 32\}$, and the hidden dimension of the decoder to 32. More details of hyperparameters are described in Appendix A.3.

### 3.2.3 DECODER

The decoder inputs the sampled latent variables $Z$ to reconstruct the adjacency matrix $A$ and gate matrix $X$. The generative process is summarized as

$$p(A|Z) = \prod_{i=1}^{K} \prod_{j=1}^{K} p(A_{ij}|z_i, z_j), \text{ with } p(A_{ij} = 1|z_i, z_j) = \text{sigmoid}_j(F_{\text{adj}}(z_i^T z_j)), \tag{3}$$

$$p(X^{\text{type}}|Z) = \prod_{i=1}^{K} p(x_i^{\text{type}}|z_i), \text{ with } p(x_i^{\text{type}}|z_i) = \text{softmax}_{l_i}(F_{\text{type}}(z_i)), \tag{4}$$

$$p(X^{\text{qubit}}|Z) = \prod_{i=1}^{K} \prod_{j=1}^{g} p(X_{ij}^{\text{qubit}}|z_i), \text{ with } p(X_{ij}^{\text{qubit}} = 1)|z_i) = \text{softmax}_j(F_{\text{qubit}}(z_i)), \tag{5}$$

where $F_{\text{adj}}, F_{\text{type}}, F_{\text{qubit}}$ are trainable linear functions and $l_i = \text{argmax}_j X_{ij}^{\text{type}}$.

### 3.2.4 OBJECTIVE FUNCTION

The weights in the encoder and decoder are optimized by maximizing the evidence lower bound $\mathcal{L}$ defined as:

$$\mathcal{L} = E_{q(Z|X,A)}[\log p(X^{\text{type}}, X^{\text{qubit}}, A|Z)] - \text{KL}[(q(Z|X, A))||p(Z)], \tag{6}$$

where $\text{KL}[q(\cdot)||p(\cdot)]$ is the Kullback-Leibler (KL) divergence between $q(\cdot)$ and $p(\cdot)$. We further adopt Gaussian prior $p(Z) = \prod_i \mathcal{N}(z_i|0, I)$. The weights are optimized by minibatch gradient descent. This work selects a batch in size of 32.

### 3.3 ARCHITECTURE SEARCH STRATEGIES

This work uses reinforcement learning and Bayesian optimization as two search strategies in the second part of our framework.

### 3.3.1 REINFORCEMENT LEARNING (RL)

After some simple trials of PPO (Schulman et al., 2017) and A2C (Huang et al., 2022), this work uses REINFORCE (Williams, 1992) as a more powerful RL algorithm for architecture search. The environment state space is a pre-trained embedding, and the agent uses a one-cell LSTM as their policy to choose an action, which is a sampled latent vector according to the distribution of the given state, and moves to the next state given the action. The energy/groud-energy is viewed as a reward for max-cut and quantum chemistry tasks. If the value is out of the [0, 1] range, it is set to the corner value 0 or 1. The circuit fidelity is a reward for the unitary design task. The reward for each step will be factorized with a predefined penalty. We use adaptive batch size as steps for each training epoch. The average reward of steps decides the number of steps for the next training epoch. Additionally, we use a linear adaptive baseline with the formula baseline $= \alpha \cdot$ baseline $+ (1 - \alpha) \cdot$ avg-reward where $\alpha$ is predefined in the range [0,1] (0.7 for the experiments of the state preparation and 0.8 for others). The number of searches is set to 1000 for each run in this work.

### 3.3.2 BAYESIAN OPTIMIZATION (BO)

As another search strategy used in this work without labeling, we employ deep networks for global optimization (DNGO) (Snoek et al., 2015) in BO. We adopt a one-layer adaptive BO regression with a basis function extracted from a feed-forward neural network and 128 units in the hidden layer to model distributions over functions. We select expected improvement (EI) (Mockus, 1977) as the acquisition function. EI takes top-5 embeddings for each training epoch with default objective 0.9. The training begins with 16 samples and adds the top-5 architectures proposed by EI in this batch for every new training epoch. Then the network is retrained for 100 epochs using the architectures selected from the updated batch. This process is iterated until the predefined search times.

## 4 EXPERIMENTAL RESULTS

To prove the effectiveness and generalized capability of our approach, we conduct experiments on three well-known applications in the field of QC including quantum state preparation, max-cut, and quantum chemistry. For each application, we first choose a simple example with 4 qubits and then select a relatively more complex example with 8 qubits. We employ the random generator to generate 100,000 circuits as our search space and all experiments so far have been performed on a noise-free simulator during the search process. Detailed settings are introduced in Appendix A.2.

In the following, we first evaluate the pre-training performance of the model for Unsupervised representation learning (§4.1) and then the QAS performance based on its pre-trained latent representations (§4.2). Additionally, some additional experiments for the pre-training process and for the QAS process are demonstrated in Appendix A.4 and Appendix A.5 respectively.

### 4.1 PRE-TRAINING PERFORMANCE

**Observation (1):** GAE and VGAE (Kipf & Welling, 2016) are two popular baselines for NAS. To explore outstanding models that can acquire superior latent representations of quantum circuit architectures, we first try these two well-known models, but quantum circuit architectures are more complex than neural network architectures, so they cannot obtain the expected results. However, the models based on GINs (Xu et al., 2019) successfully acquire valid latent representations due to its better neighbor aggregation scheme. In Table 1, we show the performance of the original model without KL divergence and the improved model with KL divergence for 4-qubit and 8-qubit circuits using four metrics: Accuracy$_{ops}$ is the reconstruction accuracy of the operation/gate matrix for the held-out test set; Accuracy$_{adj}$ is the reconstruction accuracy of the adjacency matrix for the test set; Validity refers to how often a random sample from the prior distribution can generate a valid architecture; Uniqueness refers to the ratio of unique architectures out of valid generations. The table demonstrates that the improved model outperforms the original one, because the KL term effectively regularizes the mapping from the discrete space to the continuous latent space, leading to better generative performance particularly measured by the validity. Considering its superior performance, we stick to the model for further evaluation.

Table 1: Model performance measured by reconstruction accuracy, validity, and uniqueness.

| Qubit | Model | Metric | | | |
|---|---|---|---|---|---|
| | | $Accuracy_{ops}$ | $Accuracy_{adj}$ | Validity | Uniqueness |
| 4 | $QAS_{URL}$ wo KL | 87.12 | 99.86 | 6.21 | 100 |
| 4 | $QAS_{URL}$ w KL | 99.99 | 99.99 | 86.91 | 100 |
| 8 | $QAS_{URL}$ wo KL | 99.97 | 99.97 | 20.37 | 100 |
| 8 | $QAS_{URL}$ w KL | 99.99 | 99.99 | 81.55 | 100 |

**Observation (2):** In Figure 3, we apply two popular techniques PCA (Shlens, 2014) and t-SNE (Van der Maaten & Hinton, 2008) on our pre-trained models with and without KL divergence for visualizing high-dimensional latent representation of the 4-qubit max-cut application. The results first show that using KL divergence is extremely important for acquiring a superior latent representation. Furthermore, Yang & Wu (2006) and Linderman & Steinerberger (2019) have pointed out the effectiveness of the two approaches on unsupervised clustering in addition to visualizing high-dimensional data. The figures also demonstrate that the latent representation space of quantum circuits is smooth and compact and architectures with similar performance are clustered together when our model is with KL divergence. Specifically, high-performance quantum circuit architectures are mapped more concentrated in the upper left corner of the figures. In particular, the PCA method shows extremely smooth and compact visualization results with outstanding clustering effects. Conducting QAS on such smooth latent space with clustering effects is much easier and is hence more efficient, which provides a solid foundation for our QAS algorithms.

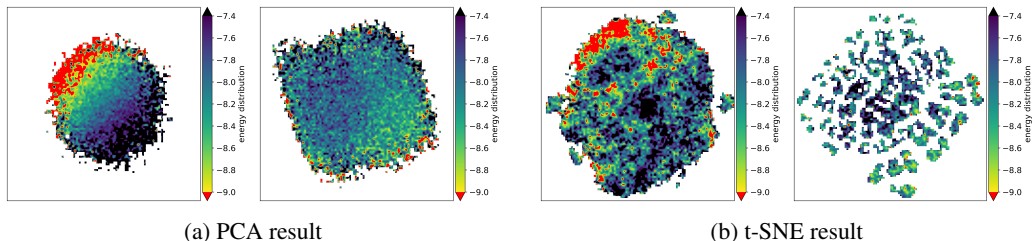

(a) PCA result  (b) t-SNE result

Figure 3: The 2D smooth visualization for the latent representation of 4-qubit max-cut with PCA and t-SNE. Color encodes the achieved energy of the randomly generated 100,000 circuits. These graphs show the energy distribution of the 100,000 circuits. Only when the achieved energy is lower than $-9.0$ $Ha$, it is encoded in red. In (a) and (b), the left one is the result of our model with KL divergence and the other is without KL divergence.

## 4.2 QUANTUM ARCHITECTURE SEARCH (QAS) PERFORMANCE

**Observation (1):** In Figure 4, we demonstrate the average reward per 100 searches for each experiment. The experimental results illustrate that the REINFORCE and BO methods effectively learn to search from the latent representation, resulting in noticeable improvements in average reward during the early stages. However, this is not achieved by Random Search. Furthermore, although the plots reveal a slightly higher variance in the average reward for REINFORCE and BO methods compared to Random Search, their overall average reward surpasses that of Random Search significantly.

**Observation (2):** In Figure 5, we demonstrate the number of candidate circuits that can be found to achieve a preset threshold by performing 1000 searches using the three search methods. The figures illustrate that the 8-qubit experiments are more complicated, so only fewer circuits that meet the requirements can be found in the search space. Additionally, Within a limited number of search iterations, the REINFORCE and BO methods can discover a greater number of candidate circuits that achieve the threshold even in the worst case, i.e. when comparing the minimal candidate quantity. Notably, their performance significantly outperforms the Random Search method, especially REINFORCE, even though the difference between the minimal and maximal candidate quantity demonstrates that REINFORCE is more sensitive to the initial value in comparison to the other

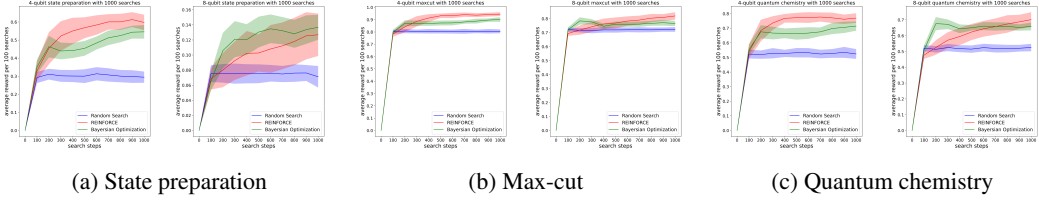

(a) State preparation      (b) Max-cut      (c) Quantum chemistry

Figure 4: Average rewards of the 6 sets of experiments. In (a), (b), and (c), the left side is the 4-qubit experiments and the right side is the 8-qubit experiments. The plots show the average reward of 50 independent runs (with different random seeds) given 1000 search queries. The shaded areas in the plots demonstrate the standard deviation of the average rewards.

two approaches. These results underscore the evident improvements and advantages introduced by QAS based on the latent representation. This approach enables the effective discovery of numerous high-performance candidate circuits while minimizing the number of searches required.

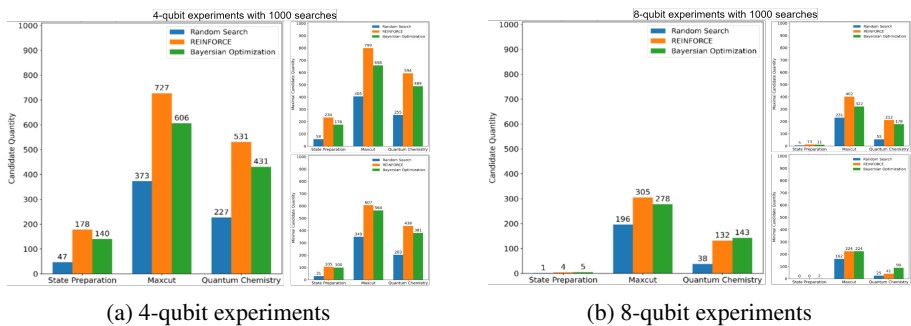

(a) 4-qubit experiments      (b) 8-qubit experiments

Figure 5: Candidate quantity of 4-qubit and 8-qubit applications. The reward thresholds of all 4-qubit experiments are 0.95, but 8-qubit experiments are more complex, so their thresholds are soft. The thresholds of the 8-qubit state preparation, max-cut, and quantum chemistry experiments are 0.75, 0.925, and 0.9 respectively. All experiments are executed for only 1000 queries, i.e. only 1000 search samples are performed from the search space of 100,000 circuits. Additionally, the left results in (a) and (b) are the average of 50 runs (using different random seeds) and the right ones correspond to the maximal and minimum candidate quantity in the 50 runs.

Table 2: Compare the QAS performance of different QAS methods for the 4-qubit state preparation.

| Method | $F_{thr}$ | $N_{lbl}$ | $N_{rest}$ | $N_{>0.95}$ | $N_{eval}$ | $N_{QAS}$ | $P_{opt}$ | $N_{QAS}/N_{eval}$ |
|---|---|---|---|---|---|---|---|---|
| GNN$_{URL}$ | 0.4 | 1000 | 18963 | 2255 | 2000 | 120 | $\approx 1.00$ | 0.0600 |
| | 0.5 | 1000 | 5182 | 913 | 2000 | 183 | $\approx 1.00$ | 0.0915 |
| GSQAS$_{URL}$ | 0.4 | 1000 | 20876 | 2380 | 2000 | 116 | $\approx 1.00$ | 0.0580 |
| | 0.5 | 1000 | 6228 | 1036 | 2000 | 168 | $\approx 1.00$ | 0.0840 |
| DQAS | - | 0 | - | - | 1000 | 0 | $\approx 0.86$ | 0 |
| QAS$_{URL\&RL}$ | - | 0 | 100000 | **4729** | 1000 | 178 | $\approx$ **1.00** | **0.1780** |

**Observation (3):** In Table 2, based on the 4-qubit state preparation and 4-qubit circuit space with 100,000 circuits, we compare other QAS methods with our approach within 1000 searches. GNN$_{URL}$ and GSQAS$_{URL}$ employ predictors in He et al. (2023b) and He et al. (2023a) respectively, but they are based on our pre-training model, QAS$_{URL\&RL}$ and DQAS denote the QAS approach with REINFORCE in this work and the method in Zhang et al. (2022) respectively. URL denotes unsupervised representation learning, $F_{thr}$ is the threshold to filter poor-performance architectures, $N_{lbl}$, $N_{rest}$ and $N_{>0.95}$ refer to the number of required labeled circuits, rest circuits after filtering and

the circuits that achieve the performance higher than 0.95 in the rest circuits respectively. $N_{eval}$ represents the number of evaluated circuits, i.e. the sum of the number of labeled and sampled circuits, $N_{QAS}$ is the number of searched candidates, and $P_{opt}$ represents the achieved optimal performance (reward in the range $[0, 1]$). The average experimental results in 50 runs show that DQAS exhibits poor search outcomes due to its sensitivity to the operation pool selection, requiring extensive sampling for limited success. The predictor-based methods and our approach can yield a substantial number of high-performance circuits with fewer samples. However, predictor-based methods rely on labeled circuits for training predictors, introducing uncertainty in filtering poor architectures, because they also filter out good architectures simultaneously. The higher $F_{thr}$ can filter more poor circuit architectures, making the proportion of good architectures in the filtered space larger and the performance of random search higher, but more well-performing architectures are sacrificed simultaneously. Despite this, our method achieves comparable performance to predictor-based methods, demonstrating higher efficiency in terms of $N_{QAS}/N_{eval}$ and $P_{opt}$ while evaluating fewer circuits. Additional experiment tables in Appendix A.5 demonstrate similar results.

**Observation (4):** In Figure 6, we present the best candidate circuits acquired by each of the three methods for every experiment. These circuits exhibit a higher likelihood of being discovered by RE-INFORCE and BO in contrast to Random Search. This observation underscores the superior search capabilities of REINFORCE and BO in navigating the large and diverse search space generated by our approach, which is based on a random generator derived from a fixed operation pool. Unlike conventional approaches that adhere to layer-wise circuit design baselines, our method excels in discovering circuits with fewer trainable parameters. This characteristic is of paramount importance when addressing real-world optimization challenges in QAS. In conclusion, our approach not only enhances the efficiency of candidate circuit discovery but also accommodates the distinct characteristics of various problem domains through a large and diverse search space.

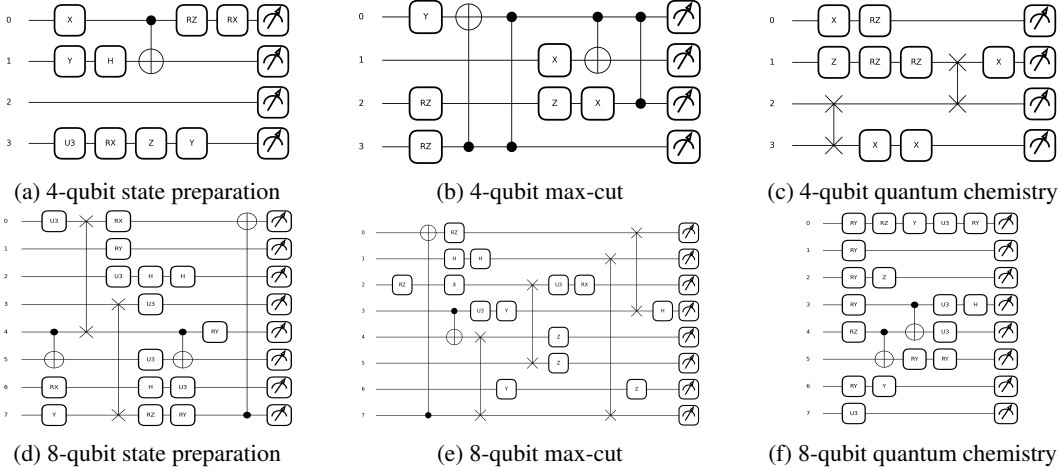

Figure 6: Best candidates of the six experiments in 50 runs.

## 5 CONCLUSION

Our framework is inspired by the method *Arch2vec* (Yan et al., 2020) to explore whether unsupervised architecture representation learning can also help QAS. By decoupling the part of unsupervised architecture representation learning from the process of QAS, we successfully eliminate the requirement for a large number of labeled circuits. Furthermore, our framework conducts the process of QAS without any predictor by directly applying the search algorithms REINFORCE and BO to the latent representation. We have demonstrated the effectiveness of our framework through various experiments. In our framework, the performance of the QAS depends on the effect of unsupervised architecture representation learning and choices of search algorithms, so we suggest that it is desirable to take a deeper investigation into architecture representation learning for QAS and designing QAS approaches using our framework with better search strategies in the latent representation.

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

# A  APPENDIX

## A.1  ENCODING LIMITATIONS AND CIRCUIT GENERATOR SETTINGS

As shown in Figure 2b, the gate matrix captures the position information of qubits that quantum gates act on, but it still has some limitations for two-qubit or multiple-qubit gates. Specifically, two-qubit control gates have a control and a target operation whose directions have an essential impact on the gate operation. For instance, asymmetric gates like `CNOT` have different unitary matrices for their original (the control operation is above the target one) and inverted forms (the target operation is above the control one), so they generate different results when acting on qubits, but it has no impact on symmetric gates such as `CZ` and `SWAP` since their original and inverted forms are same. This limitation is because the representation of qubits cannot distinguish the relative positions of the control and target qubits of two-qubit quantum gates, which destroys the uniqueness of the encoding. Therefore, we set specific generation rules for these asymmetric gates when using a random circuit generator to generate circuits. For example, only the control qubit is allowed to be above the target, i.e. inverted gates are not allowed, or like the $q_t = (q_c + 1) \ mod \ N_q$ set in our experiment where $q_t$ and $q_c$ is the corresponding qubit position of the target and control operation, and $N_q$ denotes the number of qubits. Through these rules, we sacrifice the diversity of the generated circuits to a small extent but ensure that the uniqueness of the encoding scheme is not destroyed, making our representation learning process effective.

The predefined operation pool which defines allowed gates in circuits is important for QAS as well, because a terrible operation pool such as one with no rotation gates or no control gates cannot generate numerous quantum circuits with excellent expressibility and entanglement capability. This makes the initial quantum search space poor, so it will influence our further pre-training and QAS process. Therefore, we choose some generally used quantum gates in PQCs as our operation pool $\{$X, Y, Z, H, Rx, Ry, Rz, U3, CNOT, CZ, SWAP$\}$ for the circuit generator to guarantee the generality of our quantum circuit space. Other settings of the circuit generator are summarized below:

Table 3: Description of settings predefined for the circuit generator.

| Hyperparameter | Hyperparameter explanation | Value for 4(8)-qubit experiments |
|---|---|---|
| num-qubits | the number of qubits | 4(8) |
| num-gates | the number of gates in a circuit | 10(20) |
| max-depth | the maximal depth in a circuit | 5 |
| num-circuits | required the number of circuits | $10^5$ |

## A.2  APPLICATION SETTINGS

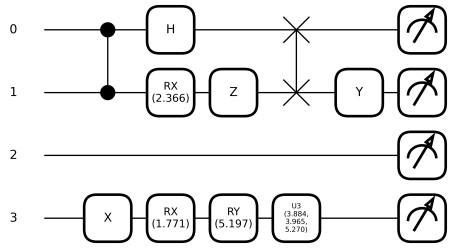

(a) The target circuit of the 4-qubit state preparation

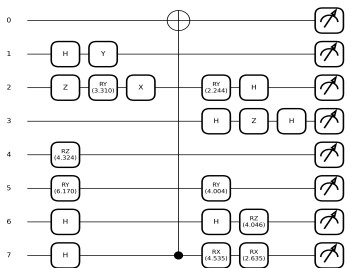

(b) The target circuit of the 8-qubit state preparation

Figure 7: The circuits used to generate the target states.

**Quantum State Preparation.** In quantum information theory, fidelity (Liang et al., 2019) is an important metric to measure the similarity of two quantum states. By introducing fidelity as the performance index, we aim to maximize the similarity of the final state density operator with a certain desired target state. We first obtain the target state by randomly generating a corresponding circuit, and then with a limited number of sample circuits, we use the search methods to search candidate circuits that can achieve a fidelity higher than a certain threshold. During the search process, the fidelity can be directly used as a normalized reward function since its range is [0, 1]. Figure 7 shows the circuits used to generate the corresponding target states.

**Max-cut Problems.** The max-cut problem (Poljak & Rendl, 1995) consists of finding a decomposition of a weighted undirected graph into two parts (not necessarily equal size) such that the sum of the weights on the edges between the parts is maximum. Over these years, the max-cut problem can be efficiently solved with quantum algorithms such as QAOA (Villalba-Diez et al., 2021) and VQE (using eigenvalues). In our work, we address the problem by deriving the Hamiltonian of the graph and using VQE to solve it. We use a simple graph with the ground state energy $-10\ Ha$ for the 4-qubit experiment and a relatively complex graph with the ground state energy $-52\ Ha$ in the case of the 8-qubit experiment. Furthermore, we convert the energy into a normalized reward function integral to the search process. The visual representations of these graphs are presented below:

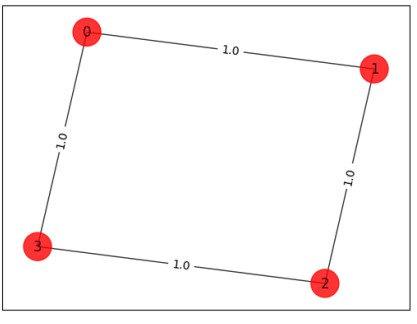 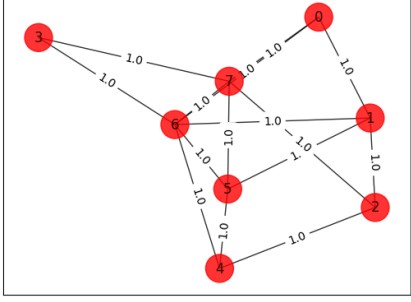

(a) The 4-qubit max-cut graph          (b) The 8-qubit max-cut graph

Figure 8: The graphs of the experiments on max-cut problems.

**Quantum Chemistry.** In the field of QC, VQE (Peruzzo et al., 2014; Tilly et al., 2022) is a hybrid quantum algorithm for quantum chemistry, quantum simulations, and optimization problems. It is used to compute the ground state energy of a Hamiltonian based on the variational principle. For the 4-qubit quantum chemistry experiment, we use the Hamiltonian of the molecule $H_2$ and its common approximate ground state energy $-1.136\ Ha$ as the optimal energy. As for the 8-qubit experiment, we consider $n = 8$ transverse field Ising model (TFIM) with the Hamiltonian as follows:

$$\boldsymbol{H} = \sum_{i=0}^{7} \sigma_z^i \sigma_z^{(i+1)\ mod\ 6} + \sigma_x^i. \tag{7}$$

We design some circuits to evaluate the ground state energy of the above Hamiltonian and get an approximate value $-10\ Ha$ as the optimal energy. According to the approximate ground state energy, we can use our methods to search candidate circuits that can achieve the energy reaching a specific threshold. In the process of searching for candidates, the energy is normalized as a reward function with the range [0, 1] to guarantee search stability.

### A.3 HYPERPARAMETERS OF PRE-TRAINING

Table 4 shows the hyperparameter settings of the pre-training model for 4-qubit and 8-qubit experiments.

Table 4: Description of hyperparameters adopted for pre-training.

| Hyperparameter | Hyperparameter explanation | Value for 4(8)-qubit experiments |
|---|---|---|
| bs | batch size | 32 |
| epochs | traning epochs | 16 (25) |
| dropout | decoder implicit regularization | 0.3 |
| normalize | input normalization | True |
| input-dim | input dimension | 2+#gates+#qubits |
| hidden-dim | dimension of hidden layer | 128 |
| dim | dimension of latent space | 16 (32) |
| hops | the number of GIN layers ($L$ in eq.2) | 5 |
| mlps | the number of MLP layers | 2 |
| latent-points | latent points for validity check | 10000 |

### A.4 LATENT DIMENSION ANALYSIS

Table 5 demonstrates a supplementary experiment for the pre-training process of the 4-qubit quantum circuit space. It can be observed from the experimental results that when the dimension of the latent space is too low, the training effect is very poor. This is because the dimension of the latent space cannot fully capture the characteristics of gate and adjacency matrices. As the dimension of the latent space increases, the model can capture the properties of the two matrices, so the performance gradually becomes better. However, continuing to increase the dimension will cause the dimension of the latent space to be too high, which will increase the difficulty of training. Simultaneously, an excessively high-dimensional potential space will increase the computational complexity, which is not what we expect, so we choose 16 as the potential feature dimension of the 4-qubit quantum circuit space. The 8-qubit one also shows similar results, but because the quantum circuit space of 8-qubit is more complex, the final selected feature dimension is 32.

Table 5: Model performance with different latent dimensions measured by reconstruction accuracy, validity, and uniqueness. The model is $\text{QAS}_{URL}$ with KL divergence and has the same training epoch 16.

| Qubit | Dimension | Metric | | | | |
|---|---|---|---|---|---|---|
| | | $\text{Accuracy}_{ops}$ | $\text{Accuracy}_{adj}$ | Validity | Uniqueness | $\text{Loss}_{avg}$ |
| 4 | 8 | 35.87 | 99.44 | 57.22 | 100 | 0.1753 |
| 4 | 16 | 99.99 | 99.99 | 86.91 | 100 | 0.0345 |
| 4 | 24 | 99.94 | 99.92 | 76.52 | 100 | 0.0371 |
| 4 | 32 | 93.92 | 99.99 | 70.02 | 100 | 0.0429 |

### A.5 SUPPLEMENT COMPARISON EXPERIMENTS OF QAS METHODS

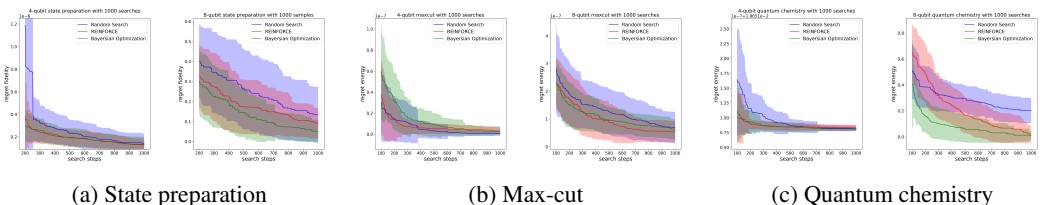

(a) State preparation   (b) Max-cut   (c) Quantum chemistry

Figure 9: Regret values of the 6 sets of experiments. In (a), (b), and (c), the left side is the 4-qubit experiments and the right side is the 8-qubit experiments. The plots show the mean regret values of 50 independent runs (with different random seeds) given 1000 search queries. The shaded areas in the plots demonstrate the standard deviation of the regret value.

Figure 9 presents regret values for each experiment, representing the difference between actual results and their optimal counterparts. The plots intuitively illustrate that, with the exception of the 4-qubit max-cut experiment, the REINFORCE and BO algorithms excel in the search for quantum architectures, more approaching optimal values. This advantage becomes particularly pronounced in 8-qubit complex applications. Furthermore, our results demonstrate that both methods consistently yield smaller average regret variances than Random Search when searching for circuit architectures that closely approximate optimal values, underscoring their stability and reliability.

Table 6 and 7 are additional experiments max-cut and quantum chemistry for Table 2. The experimental results show the same conclusions.

Table 6: Compare the QAS performance of different QAS methods for the 4-qubit max-cut.

| Method | $F_{thr}$ | $N_{lbl}$ | $N_{rest}$ | $N_{>0.95}$ | $N_{eval}$ | $N_{QAS}$ | $P_{opt}$ | $N_{QAS}/N_{eval}$ |
|---|---|---|---|---|---|---|---|---|
| GNN$_{URL}$ | 0.85 | 1000 | 31338 | 19186 | 2000 | 612 | $\approx 1.00$ | 0.3060 |
| | 0.9 | 1000 | 9778 | 7234 | 2000 | 743 | $\approx 1.00$ | 0.3715 |
| GSQAS$_{URL}$ | 0.85 | 1000 | 32012 | 18809 | 2000 | 588 | $\approx 1.00$ | 0.2940 |
| | 0.9 | 1000 | 13935 | 9479 | 2000 | 685 | $\approx 1.00$ | 0.3425 |
| QAS$_{URL\&RL}$ | - | 0 | 100000 | **37709** | 1000 | 727 | $\approx$ **1.00** | **0.7270** |

Table 7: Compare the QAS performance of different QAS methods for the 4-qubit quantum chemistry.

| Method | $F_{thr}$ | $N_{lbl}$ | $N_{rest}$ | $N_{>0.95}$ | $N_{eval}$ | $N_{QAS}$ | $P_{opt}$ | $N_{QAS}/N_{eval}$ |
|---|---|---|---|---|---|---|---|---|
| GNN$_{URL}$ | 0.6 | 1000 | 28322 | 11277 | 2000 | 376 | $\approx 0.98$ | 0.1880 |
| | 0.675 | 1000 | 12303 | 5944 | 2000 | 458 | $\approx 0.98$ | 0.2290 |
| GSQAS$_{URL}$ | 0.6 | 1000 | 27201 | 10820 | 2000 | 387 | $\approx 0.98$ | 0.1935 |
| | 0.675 | 1000 | 12608 | 5966 | 2000 | 452 | $\approx 0.98$ | 0.2260 |
| QAS$_{URL\&RL}$ | - | 0 | 100000 | **22514** | 1000 | 531 | $\approx$ **0.98** | **0.5310** |

