# OpenReview forum: "Quantum Architecture Search with Unsupervised Representation Learning"
_ICLR.cc/2024/Conference — ICLR 2024 Conference Withdrawn Submission_

### Official Review · Reviewer_asUZ · 2023-10-18

**Soundness:** 2 fair
**Presentation:** 3 good
**Contribution:** 2 fair
**Rating:** 5
**Confidence:** 3

**Summary:**

The paper proposes to learn unsupervised quantum circuit representations with a variational graph autoencoder (VGAE) approach to support quantum architecture search. Smooth and compact latent representations of circuit can be learned, and well-performing candidate circuits can be quickly identified with REINFORCE and BO on various downstream tasks.

The main contribution of this work is to effectively decouple the unsupervised architecture
representation learning from QAS, so that predictor-free QAS algorithms could be applied without relying on labels of quantum circuit architectures.

**Strengths:**

- The paper targets the frontier research topic of QAS and proposes a theoretically sound idea of unsupervised learning of quantum circuit representations.

- The proposed algorithm is well illustrated in Figure 1 and easy to follow.

- Comprehensive experiments are carried out to show the performance of the proposed model, both in terms of the quality of pre-trained latent quantum circuit representation and in terms of the performance of QAS algorithms on top of it.

- The article presents an inclusive overview and insightful discussion of related works.

**Weaknesses:**

- The contribution of the work appears incremental to me, as the main novelty of the paper sits in the employment of an existing VGAE approach for encoding quantum circuits.

- It seems that there is a logic mismatch between the main claim and the designed experiment to prove it.  In order to show the usefulness of the unsupervised quantum circuit encoding scheme on QAS, one should compare it with other schemes followed by the same QAS algorithms, and include strong predictor-based QAS model, i.e. He et al. (2023a), as references to the absolute performance of the proposed approach. Unfortunately, they are missing from the experiments.

- The experiment is weak in the models under comparison: only one baseline system is included in both experiments, and they are not strong enough to reach a convincing conclusion, especially the second.

**Questions:**

The proposed idea of the paper is interesting and technically sound, but the contributions are limited due to the weaknesses in the experiment design. I don't mind changing my mind if spot-on clarifications could be given in regard to the comments above as well as the following questions:

- Are there underlying reasons why the proposed VGAE have the clustering effect? What do the PCA and t-SNE result look like for the model without the KL divergence?

- How does the proposed approach work in comparison to predictor-based QAS algorithms?

- It seems that quantum circuits with a greater depth are more likely to suffer from the Barren Plateau problem. Can this factor be rendered in the pre-training model or in the design of rewards in the QAS algorithm?

---

> ### Author Response · Authors · 2023-11-21
> **Official Comment by Authors**
>
> We are very grateful for your constructive feedback, here are our comments:
> > **W1:**
> > The contribution of the work appears incremental to me, as the main novelty of the paper sits in the employment of an existing VGAE approach for encoding quantum circuits.
>
> Many existing QAS methods are inspired by and modified from NAS methods. Compared with classical neural network architectures, quantum architectures have certain similarities, but are also more complex. Therefore, it is challenging in itself for us to explore NAS algorithms suitable for QAS and undergo a series of improvements suitable for quantum architectures. We agree with you that our innovation is incremental, but our innovation goes beyond using VGAE to encode quantum circuits. It also involves us no longer relying on predictors, but designing search algorithms directly based on latent representations. Such a completely decoupled structure makes our architecture not dependent on any downstream tasks and has better scalability.
>
> > **W2:**
> > It seems that there is a logic mismatch between the main claim and the designed experiment to prove it. In order to show the usefulness of the unsupervised quantum circuit encoding scheme on QAS, one should compare it with other schemes followed by the same QAS algorithms, and include strong predictor-based QAS model, i.e. He et al. (2023a), as references to the absolute performance of the proposed approach. Unfortunately, they are missing from the experiments.
>
> Do you mean we should add comparisons about the pre-trained models and latent representations between different QAS methods? We mentioned in the manuscript that we use the same encoding scheme as He et al. (2023a), but QAS does not have unified baselines such as NAS-Bench101, NAS-Bench201, and NAS-Bench301 like NAS, so comparison requires a large number of repeated and reproducted experiments, i.e. evaluating a huge number of quantum circuits to get their performance to conduct visualisation. Quantum experiments are much more time-consuming than classical experiments, which is not easy. If you mean that we should add experiments to compare the performance of these algorithms based on the same encoding pattern (e.g. He et al. (2023a)), we agree with that. We are indeed missing these comparisons, so we conducted additional experiments and added Table 2, 6 and 7 in **pages 8 and 17** for comparison to make our statements convincing.
>
> > **W3:**
> > The experiment is weak in the models under comparison: only one baseline system is included in both experiments, and they are not strong enough to reach a convincing conclusion, especially the second.
>
> We indeed missed some comparisons. Therefore, we added some comparison experiments with other QAS methods in the revised version to make our statements convincing. You can find Table 2, 6, and 7 in **pages 8 and 17**. And as you said, the second (figures used to show regret reward) is not strong enough to reach a convincing conclusion, so we move them to the appendix and only keep the first one (figures to show average reward).

---

> ### Author Response · Authors · 2023-11-21
> **Answer to Questions**
>
> > **Q1:**
> > Are there underlying reasons why the proposed VGAE have the clustering effect? What do the PCA and t-SNE result look like for the model without the KL divergence?
>
> This question is interesting and important. What we want to emphasize is that we decouple the latent representation learning process of the circuit structure and the QAS process, so our VGAE with the GIN model directly learns the relationship on the structural representation. This makes sense because different downstream tasks are likely to rely on different quantum circuit structures, so decoupling allows the underlying representation to be adapted to different downstream tasks. When it comes to the potential reasons why VGAE has a clustering effect, we think it is because similar circuit structures may have similar performance, and the GIN model can learn the dependencies between nodes and previous and subsequent nodes, so it can be very good. Capture the connection relationships of nodes in the model and the similarities between models. In fact, the predictor-base method also shows this to a certain extent, because if similar structures do not have similar performance at all, then the predictor cannot learn the relationship between latent features and performance. Of course, quantum circuit architecture is much more complex than classical neural networks, so we can only infer the above conclusions from existing work and our experiments. More experiments and work are needed to fully verify the above conclusions.
>
> The PCA and t-SNE results for the model without the KL divergence are added in Figure 3 (**page 7**) of the revised version. The results show that when the model is without KL divergence, it cannot reflect the clustering effect, and even the latent representation may become less compact, smooth, and continuous.
>
> > **Q2:**
> > How does the proposed approach work in comparison to predictor-based QAS algorithms?
>
> The predictor-based QAS method is also motivated by the predictor-based NAS method such as "Neural predictor for neural architecture search" (https://arxiv.org/pdf/1912.00848.pdf). They use their models to learn a latent representation and randomly train numerous quantum circuits to pre-train a predictor with the circuits' latent features and performance. After getting the predictor, they use the predictor to filter the bad-performance circuits, but many well-performing circuits are filtered. However, the predictor indeed learns something, so more bad-performance circuits are filtered. After the search space is filtered, the ratio of the well-performing circuits becomes larger, and using random search can reevaluate the rest circuits and get a large number of good candidate circuits. However, our proposed model directly uses REINFORCE and Bayesian Optimization to search on the latent representation, and evaluate the circuit performance, return the performance to optimize the agent or the network. Therefore, our model can eliminate the need for predictors. Additionally, the predictor needs some labeled circuits and also causes uncertainty, because it needs to learn the relationship between latent features and performance, but the relationship between latent features and performance is not easy to learn and not absolute, so the predictor also filters many good circuits. The predictor also depends on different downstream tasks, so different downstream tasks need different labels. All these problems are solved by our proposed methods. In order to show the performance of our model is better than the predictor-based method, we added some tables in the revised version to compare them.
>
> > **Q3:**
> > It seems that quantum circuits with a greater depth are more likely to suffer from the Barren Plateau problem. Can this factor be rendered in the pre-training model or in the design of rewards in the QAS algorithm?
>
> This is an interesting question. BP is a very important problem in quantum computing.
>
> Firstly, when using the random circuit generator to generate circuits, we predefined a maximal depth to avoid so deep circuits.
>
> Secondly, we think the adjacency matrix reflects the depth of circuits to some extent, but it is not direct, so we are not sure if the depth is rendered in the pre-training model. We can directly add depth as another information in the pre-training model like supervised models adding performance information, but it may result in the latent representation not being smooth and continuous.
>
> Thirdly, we can confirm that the depth factor can be a penalty factor for the design of rewards in the QAS algorithms, such as $reward_{new} = reward * e^{(-c * depth)}$ where $c$ is a constant factor, and this has an impact on the QAS process.

---

### Official Review · Reviewer_QFKe · 2023-10-23

**Soundness:** 3 good
**Presentation:** 3 good
**Contribution:** 2 fair
**Rating:** 5
**Confidence:** 3

**Summary:**

This article proposes a quantum architecture search (QAS) framework with unsupervised representation learning. The framework consists of two parts: an autoencoder that learns a latent representation of quantum circuit architectures without any labels, and a search algorithm that directly optimizes the latent representation using reinforcement learning or Bayesian optimization. The framework aims to improve the efficiency and generality of QAS by avoiding the need for a large number of labeled circuits and a predictor. The authors demonstrate the effectiveness of their framework on three applications: fidelity of quantum states, max-cut, and quantum chemistry.

**Strengths:**

- The article is well-written and presents an interesting unsupervised approach to QAS. It applies reinforcement learning and Bayesian optimization to directly search for the latent representation, avoiding the need for a predictor and a large number of labeled circuits.

- The proposed method decouples the representation learning from the search process, making QAS more efficient. They also visualize and analyze the learned latent representation and show that it is smooth and clustered.

**Weaknesses:**

**The lack of comparison with key relevant methods.**  The paper claims that sampling-based QAS methods have inefficient performance and high evaluation costs, but there are no comparative experiments to verify this argument. In fact, the paper's experiments were not compared with any other method, making it hard to see what performance improvements are being made.

**Questions:**

1. **On the scalability and robustness**: How does the performance perform when the proposed method compares with existing QAS methods in terms of search efficiency, scalability, and robustness to noise?  This may be important to reflect the applicability and generalization to real-world quantum devices.

2. **On the choice of operation pool**: How did you choose the operation pool for the quantum circuits? Could the choice of operation pool introduce bias or limit the diversity of the generated circuits?

---

> ### Author Response · Authors · 2023-11-21
> **Official Comment by Authors**
>
> ### W:
> **The lack of comparison with key relevant methods.**
> The paper claims that sampling-based QAS methods have inefficient performance and high evaluation costs, but there are no comparative experiments to verify this argument. In fact, the paper's experiments were not compared with any other method, making it hard to see what performance improvements are being made.
>
> **Answer**:
> We thank the reviewer for pointing out this issue. We conduct experiments and add a table 2, 6, 7 (**page 8, 17**) for comparing different QAS methods.
>
> In Table 2 (**page 8**), based on the 4-qubit state preparation and 4-qubit circuit space with 100,000 circuits, we compare other QAS methods with our approach within 1000 searches. The predictor-based methods and our approach can yield a substantial number of high-performance circuits with fewer samples. However, predictor-based methods rely on labeled circuits for training predictors, introducing uncertainty in filtering poor architectures, because they also filter out good architectures simultaneously. The higher $F_{thr}$ can filter more poor circuit architectures, making the proportion of good architectures in the filtered space larger and the performance of random search higher, but more well-performing architectures are sacrificed simultaneously. Despite this, our method achieves comparable performance to predictor-based methods, demonstrating higher efficiency in terms of $N_{QAS}/N_{eval}$ and $P_{opt}$ while evaluating fewer circuits.
>
> The sampling of our algorithm provides efficiency compared with another sample-based algorithm DQAS. Our algorithm can search more well-performing structures with much less sampling. But DQAS is noted by learning circuit weights and architecture simultaneously. DQAS has an operation pool and a probability model. The operation pool do not include 100000 circuits but only some circuit component gates. DQAS uses the the architecture parameter $\alpha_{ij}$ and equation
>         $\begin{align}
>             \mathcal{L}=\sum_{k\sim P(k,\alpha)}\frac{P(k,\alpha)}{\sum_{k^{'}\sim P(k,\alpha)}P(k^{'},\alpha)}L_k(\theta)
>         \end{align}
>         $, where
>         $
>         \begin{align}
>             P(k,\alpha)=\prod_i^p \frac{e^{\alpha_{ij}}}{\sum_k e^{\alpha_{ik}}}
>         \end{align}$ to transform the search space from discrete into continuous and to approximate the architecture distribution. The $k$ here denotes the sampled structure from the probability model $P(k,\alpha)$.
>
> According to the distribution, it samples a batch of candidate architectures for each epoch. The total number of samples depends on the number of training epochs and batch size of samples per epoch. DQAS samples a batch of candidate architectures for each training epoch and conducts gradient descent for each architecture parameter, and weights iteratively. This training process couples the search space and evaluation and thus needs a lot of training epochs (time-consuming). However, our method uses unsupervised representation learning to first embed the discrete circuit pool in a continuous latent space, which does not rely on downstream tasks. Additionally, the performance of DQAS relies more on the selection of the operation pool compared with our method.

---

> ### Author Response · Authors · 2023-11-21
> **Answer to Question**
>
> ### Questions:
> **Q1**:
> On the scalability and robustness: How does the performance perform when the proposed method compares with existing QAS methods in terms of search efficiency, scalability, and robustness to noise? This may be important to reflect the applicability and generalization to real-world quantum devices.
>
> **Answer**:
> Thanks for these questions. You are right, we added tables (Table 2, 6, 7 in **page 8 and 17**) in the revised version to compare the performance of part of the existing QAS methods.
>
> Predictor-based QAS algorithms need to label a set of architectures for pre-training, so they suffer from the label cost and label accuracy problem. They would filter out a lot of bad structures by learning the latent space, but they also filter out some well-performing architectures, which leads to unstable search results in the search process and missing some good circuits.
>
> The learning constraints of predictor-based algorithms would also limit the scalability of different search tasks. However, our algorithm can learn a latent space without any labeling. It can learn a latent space without any missed structures owing to filtering. Since we decouple the unsupervised learning and search method, it provides a more flexible way to combine different latent space and learning methods with different search algorithms.
>
> The sampling of our algorithm provides efficiency compared with another sampling-based algorithm DQAS. Our algorithm can search more well-performing structures with much less sampling. But the efficiency of DQAS is noted by learning structure and weights simultaneously. If there could be a benchmark dataset like NAS-Bench101 for comparing different architecture search algorithms, the performance distinctions of different algorithms would be shown directly.
>
> Regarding the robustness to noise, you are right, our algorithm lacks the analysis regarding robustness to noise. We tested our algorithm on a noise-free simulator as well as many other unsupervised learning-based QAS algorithms, but this is still a very interesting direction in future work.
>
> **Q2**:
> On the choice of operation pool: How did you choose the operation pool for the quantum circuits? Could the choice of operation pool introduce bias or limit the diversity of the generated circuits?
>
> **Answer**:
> Thanks for this question. The operation pool is created by selecting basic quantum gates for creating a quantum circuit. We select common logic single-qubit gates Pauli X, Y, Z, and Hardmard gate, basic trainable gates Rx, Ry, Rz and U3 to create arbitrary unitary operation and two-qubit gates CNOT, CZ, and SWAP for entanglement, since these operations are generally used quantum gates in parameterized or variational quantum circuits. Therefore, using the operation pool at least can guarantee the generality of our quantum circuit space.
>
> The predefined operation pool which defines allowed gates in circuits is important for QAS as well because a terrible operation pool such as one with no rotation gates or no control gates cannot generate numerous quantum circuits with excellent expressibility and entanglement capability. This makes the initial quantum search space poor so it will influence our further pre-training and QAS process. When the number of allowed gates in the operation pool is so small, it limits the diversity of the generated circuits. In summary, the choice of the operation pool can introduce bias and limit the diversity of the generated circuits.

---

### Official Review · Reviewer_o6Zq · 2023-10-29

**Soundness:** 3 good
**Presentation:** 4 excellent
**Contribution:** 2 fair
**Rating:** 5
**Confidence:** 4

**Summary:**

This manuscript introduces a framework designed to address the challenge of quantum architecture search by representation learning. The authors utilize an autoencoder-based model to construct unsupervised representations for circuit architectures and apply them to different downstream tasks. Furthermore, the authors show the model's effectiveness by employing it in different types of quantum tasks.

**Strengths:**

- This manuscript employs an unsupervised framework for constructing representations of circuit architectures, eliminating the need for an extra dataset of labeled circuits during model training.

- The proposed method demonstrates versatility, as evidenced by its effectiveness across various types of tasks, as illustrated by the numerical experiments.

- The presentation of the paper is excellent. The clustering figure vividly elucidates the properties of the latent representations.

**Weaknesses:**

This work is essentially an application of Arch2vec to quantum circuits and representing a quantum circuit with a Directed Acyclic Graph (DAG) is also a preexisting method. Therefore, the primary weakness of this study lies in its lack of novelty. I suggest the authors perform a more in-depth analysis of the distinctions between quantum circuits and classical neural networks.

**Questions:**

- I'm curious about how the model ensures that an arbitrary high-dimensional representation accurately corresponds to a specific circuit architecture. In essence, I'd like to understand if there is a possibility that the decoder's output may not yield a valid gate matrix or adjacency matrix.
- It appears that the size of the gate matrix and adjacency matrix depend on the number of gates. Could the authors clarify how they determine these sizes before the entire process?
- What criteria are used for selecting the initial values of the latent representation in the search process? It would be valuable if the authors could offer a worst-case analysis in this regard.
- How is the dimension of the latent representations determined? I suggest that the authors consider conducting additional experiments to demonstrate the impact of dimensions in the latent representation space.

---

> ### Author Response · Authors · 2023-11-21
> **Official Comment by Authors**
>
> ### W:
> This work is essentially an application of Arch2vec to quantum circuits and representing a quantum circuit with a Directed Acyclic Graph (DAG) is also a preexisting method. Therefore, the primary weakness of this study lies in its lack of novelty. I suggest the authors perform a more in-depth analysis of the distinctions between quantum circuits and classical neural networks.
>
> **Answer**:
> We respectfully disagree with the novelty issue. Although many QAS algorithms were inspired by classical NAS algorithms, there are significant distinctions between classical neural networks and quantum circuits (shown in the following points).
>
> Due to these distinctions, there is no guarantee for a simple application. Our work does not trivially apply Arch2vec to quantum circuits, it has its own intuition and insight, based on such a unique design. In quantum machine learning, the training of a quantum circuit is very time-consuming. We want to evaluate the circuit as few as possible by searching.
>
> Generally, we want to search circuits with less labeling and less sampling by keeping circuit candidates as many and diverse as possible for different tasks. Thus, we develop the predictor-free framework and use RL and BO for tests, which do not need any labeling for searching. Since most QAS methods couple the search space and evaluation by searching, we decouple the latent space learning process with the search method; this unique design could provide more flexibility and scalability for searching and shows its novelty in QAS with an unsupervised learning method.
>
> **Distinction**:
>
> 1. **Data representation**: Classical neural networks represent data in a classical way, with all information encoded as classical bits, while quantum neural networks have the characteristics of quantum states and can represent information in a quantum superposition, allowing multiple inputs to be processed in parallel.
> 2. **Computational units and processes**: Classical neural networks use classical computational units, such as neurons or perceptrons, whose states, weight calculations, and activation functions are all classical. They perform traditional computational operations such as addition and multiplication. However, quantum neural networks use qubits as computing units, and their computing process involves quantum gate operations, which can be adjusted through learnable parameters in a PQC. The entire process follows the principles of quantum mechanics.
> 3. **Superposition and entanglement properties**: Classical neural networks have superposition properties that are easy to understand and explain, that is, adding more neurons or layers to the network may enhance its representation ability, while the superposition properties of quantum neural networks are more complex and involve quantum entanglement and quantum properties of interactions.

---

> > ### Comment · Reviewer_o6Zq · 2023-11-22
> >
> > I appreciate the authors' efforts to answer my questions. However, I will maintain my rating.
> >
> > I'm familiar with quantum concepts, and I know that there are various differences between classical neural networks and quantum circuits. However, I believe that the proposed model in this work doesn't seem to leverage these properties of quantum circuits. Additionally, I'm a bit confused about the authors' assertion that they developed the predictor-free framework and utilized RL and BO for tests. From my viewpoint, this appears to align closely with the contributions made by Arch2vec.

---

> ### Author Response · Authors · 2023-11-21
> **Answer to Question**
>
> ### Questions:
> **Q1**:
> I'm curious about how the model ensures that an arbitrary high-dimensional representation accurately corresponds to a specific circuit architecture. In essence, I'd like to understand if there is a possibility that the decoder's output may not yield a valid gate matrix or adjacency matrix.
>
> **Answer**:
> Thanks for your question. Yes, it is possible that the decoder gives back an invalid gate or adjacency matrix. For example, the START or END gate appears in the middle of reconstructed graphs. The KL divergence used in this work can help us to reduce the number of invalid structures. However, based on our training experience, a lower loss can not ensure a larger validity of the reconstructed matrices. We have used the validity of 10000 latent points to check if the pre-trained model generates valid structures with high probability. These values of validity are documented and compared among different models in Tables 1 (**page 7**) and 5 (**page 16**).
>
> **Q2**:
> It appears that the size of the gate matrix and adjacency matrix depend on the number of gates. Could the authors clarify how they determine these sizes before the entire process?
>
> **Answer**:
> You are right. The number of gates is predefined when using the random circuit generator to generate circuits. In other words, circuits in the search space have the same number of quantum gates, but different depths and gates. We mention this point when introducing our framework in Figure 1, but not so detailed, so we add some further descriptions in Appendix A.1 (**page 14**). When the number of gates is determined, then the sizes of the gate and adjacency matrices are determined.
>
> **Q3**:
> What criteria are used for selecting the initial values of the latent representation in the search process? It would be valuable if the authors could offer a worst-case analysis in this regard.
>
> **Answer**:
> In the process of QAS, the initial values are random, so our experiments are the average results or summary results of 50 runs. In the modified version, in Figures 4 and 5 we show the worst-case results to a certain extent. Figure 4 shows the variance, and Figure 5 shows the minimum number of candidate circuits that can be found in 50 runs. These results show that when the initial value is not good, i.e. far away from the high-performance circuit cluster, the performance of the search will decrease, but it is still better than random search.
>
> **Q4**:
> How is the dimension of the latent representations determined? I suggest that the authors consider conducting additional experiments to demonstrate the impact of dimensions in the latent representation space.
>
> **Answer**:
> Thanks for your questions and suggestions, the dimensions of our latent features were determined experimentally. Experiments show that the latent space dimension that is too small is not enough to capture the characteristics of the gate and adjacency matrix, so the pre-training performance is very poor (check the 4 metrics mentioned in Table 1).
>
> As the dimension of the latent space increases, the model can capture the properties of the two matrices, so the performance gradually becomes better. However, continuing to increase the dimension will cause the dimension of the latent space to be too high, which will increase the difficulty of training. Simultaneously, an excessively high-dimensional potential space will increase the computational complexity, which is not what we expect, so we choose 16 as the potential feature dimension of the 4-qubit quantum circuit space. The 8-qubit one also shows similar results, but because the quantum circuit space of 8-qubit is more complex, the final selected feature dimension is 32. We add the experiment results in Table 4 to Appendix A.4 of the revised version (**page 16**).

---

> ### Author Response · Authors · 2023-11-22
> **Official Comment by Authors**
>
> Thank you very much for your comment.
>
> I'm sorry that my statement confused you. You can find that we have indeed stated that our algorithm was inspired by arch2vec, and we listed our contributions in our original paper, and briefly described our motivation in our reply.
>
> Quantum circuits are different from classical neural networks: a quantum circuit needs to convert classical inputs into a quantum state and obtain outputs through physical observations rather than binary calculations, which is essentially different from classical neural networks. Using complex numbers instead of real numbers as weights increases the difference between the two models. This difference is also reflected in the quantum circuit's depth, width, and number of gates. They represent two ways of describing the world.
>
> Innovation does not just focus on one's field but requires inspiration from different fields. You can also find that many quantum architecture search algorithms are inspired by classical architecture search. For example, [DQAS](https://arxiv.org/abs/2109.15273)[1] is inspired by [DARTS](https://arxiv.org/abs/1806.09055)[2], and [QNEAT](https://arxiv.org/abs/2304.06981)[3] is inspired by [NEAT](https://nn.cs.utexas.edu/downloads/papers/stanley.ec02.pdf)[4]. [quantumDARTS](https://openreview.net/forum?id=jGYxcXSg8C)[5] is inspired by DARTS and [AGNAS](https://proceedings.mlr.press/v162/sun22a/sun22a.pdf)[6]. The same goes for our algorithms.
>
> Many thanks in advance, if you can reconsider the contribution of our article to quantum architecture search, or provide us with more suggestions for modifications.
>
> [1] S.-X. Zhang, C.-Y. Hsieh, S. Zhang, and H. Yao, “Differentiable
> quantum architecture search,” arXiv preprint arXiv:2010.08561, 2020
>
> [2]H. Liu, K. Simonyan, and Y. Yang, “Darts: Differentiable architecture
> search,” arXiv preprint arXiv:1806.09055, 2018.
>
> [3]A. Giovagnoli, V. Tresp, Y. Ma, and M. Schubert, “Qneat: Natural
> evolution of variational quantum circuit architecture,” in Proceedings of
> the Companion Conference on Genetic and Evolutionary Computation,
> pp. 647–650, 2023
>
> [4]K. O. Stanley and R. Miikkulainen, “Evolving neural networks through
> augmenting topologies,” Evolutionary computation, vol. 10, no. 2,
> pp. 99–127, 2002.
>
> [5]W. Wu, G. Yan, X. Lu, K. Pan, and J. Yan, “Quantumdarts: Differentiable
> quantum architecture search for variational quantum algorithms,” 2023.
>
> [6]Z. Sun, Y. Hu, S. Lu, L. Yang, J. Mei, Y. Han, and X. Li, “Agnas:
> Attention-guided micro and macro-architecture search,” in International
> Conference on Machine Learning, pp. 20777–20789, PMLR, 2022

---

### Official Review · Reviewer_EmZc · 2023-10-31

**Soundness:** 3 good
**Presentation:** 2 fair
**Contribution:** 2 fair
**Rating:** 5
**Confidence:** 3

**Summary:**

The paper presents a framework for quantum architecture search (QAS) that has a decoupled unsupervised architecture representation learning procedure. Based on the framework, the authors propose to use graph isomorphism networks to learn the representation of the variational quantum circuits, which enables the visualization of effective circuits. Then reinforcement learning and Bayesian optimization are applied to search for performing circuits from the representation of concrete tasks. The method is demonstrated on three typical tasks of variational quantum algorithms and generates variational circuits that perform well specifically on these tasks.

**Strengths:**

The framework decouples the unsupervised learning of circuit representation, which provides insights on what are the essential characteristics of variational circuits that make them perform well. This feature also dismisses the data labeling and the predictor training procedures of the prior frameworks for QAS.

Multiple experiments are conducted to demonstrate the feasibility of this new framework to support the claims.

**Weaknesses:**

The experiments are only conducted inside the new framework with alternated quantum architecture search methods. Further comparison to other QAS methods is necessary to understand the limit of the proposed method, corresponding to its drop of some hard subroutines of other methods.

I think the aggregated circuit representations can have more intuitive explanations to understand. From my observations of the circuits displayed in Figure 3, it seems that having more parameters on each qubit and having several entangling generating gates to forge entanglement among the qubits are the key to the performance of the circuit and is also the key feature in the latent representation learning. I wonder if there are other features that are essential, and whether they can be captured by the graph representation of the circuit.

The evaluation of the search procedure is also limited. Only three methods are compared. I keep wondering what are the good design principles for the methods of quantum architecture search.

Some minor points:
* Why in the gate matrix representation in Figure 2, CNOT gate is not directed? I.e., the order of q1 and q2 matters for the gate.
* The notations around (2) are inconsistent. MLP^(s) and MLP(M^(S)) seem to be the same thing.
* The naming of “fidelity of quantum states” is not conventional. It’s better to use the name “state preparation”.
* The details in the training of the GINs learning of circuit representations are missing in the appendix.

**Questions:**

See the above weaknesses.

---

> ### Author Response · Authors · 2023-11-21
> **Official Comment by Authors**
>
> Thank you for the detailed constructive comments and questions, here are our replies to your comments.
>
> > **W1:**
> > The experiments are only conducted inside the new framework with alternated quantum architecture search methods. Further comparison to other QAS methods is necessary to understand the limit of the proposed method, corresponding to its drop of some hard subroutines of other methods.
>
> Thanks for your pointed weakness, we added tables (Table 2 on **(page 8)**, additional Table 6 and Table 7 in the appendix on **(page 17)** in the revised version to compare some different QAS methods. The results in the tables especially $N_{QAS}/N_{eval}$ can show the efficiency of our proposed method.
>
> > **W2:**
> > I think the aggregated circuit representations can have more intuitive explanations to understand. From my observations of the circuits displayed in Figure 3, it seems that having more parameters on each qubit and having several entangling generating gates to forge entanglement among the qubits are the key to the performance of the circuit and is also the key feature in the latent representation learning. I wonder if there are other features that are essential, and whether they can be captured by the graph representation of the circuit.
>
> Thanks for your questions. Yes, the trainable gates and entanglement are essential for well-performing circuits. However, two-qubit gates like CNOT gates have much larger costs on real hardware compared with other single-qubit gates. More parameters indicate more often more trainable gates and deeper circuits, which could lead to the BP problem. Therefore, we expect to find diverse circuit architectures that have a certain number of trainable and entanglement gates, but keep the number of them in a reasonable range. It is because the number of trainable and entanglement gates plays an important role in the performance of circuits, but it does not mean that more trainable and entanglement gates will bring more improved performance, especially on real hardware. On real hardware, there are so many constraints, such as the noise, and BP problem (circuit depth). Besides the trainable and entanglement gates, some other information about architectures can be captured to some extent, such as the depth of the circuits, and interaction between qubits, but something that needs to be quantified like noise can't be rendered.
>
> > **W3:**
> > The evaluation of the search procedure is also limited. Only three methods are compared. I keep wondering what are the good design principles for the methods of quantum architecture search.
>
> Thanks for your questions, you are right.  Due to hardware limitations, the training of a quantum circuit is very time-consuming. We want to evaluate the circuit as few as possible by searching. Generally, we want to search circuits with fewer evaluations and less sampling, meanwhile, we want to keep circuit candidates as many and diverse as possible for different tasks. In Table 2 (page 8), based on the 4-qubit state preparation and 4-qubit circuit space with 100,000 circuits, we compare other QAS methods with our approach within 1000 searches. The average experimental results in 50 runs show that DQAS exhibits poor search outcomes due to its sensitivity to the operation pool selection, requiring extensive sampling for limited success. The predictor-based methods and our approach can yield a substantial number of high-performance circuits with fewer samples. However, predictor-based methods rely on labeled circuits for training predictors, introducing uncertainty in filtering poor architectures, because they also filter out good architectures simultaneously. The higher $F_{threshold}$ can filter more poor circuit architectures, making the proportion of good architectures in the filtered space larger and the performance of random search higher, but more well-performing architectures are sacrificed simultaneously. Despite this, our method achieves comparable performance to predictor-based methods, demonstrating higher efficiency in terms of $N_{QAS}/N_{eval}$ and $P_{opt}$ while evaluating fewer circuits.

---

> ### Author Response · Authors · 2023-11-21
> **Answer to Minor Points**
>
> > **Q1:**
> > Why in the gate matrix representation in Figure 2, CNOT gate is not directed? I.e., the order of q1 and q2 matters for the gate.
>
> You are right to be concerned, due to page limits we have not mentioned encoding limitations in this paper. The limitations are already mentioned in the paper [GSQAS](https://doi.org/10.48550/arXiv.2303.12381). This limitation is because the representation of qubits cannot distinguish the relative positions of the control and target of two-qubit quantum gates. This has no impact on symmetric two-qubit quantum gates such as CZ and SWAP, but it will on asymmetric gates such as CNOT. This destroys the uniqueness of the encoding, so to avoid such problems, we set certain generation rules for these asymmetric gates when using a random circuit generator to generate circuits. For example, only the control bit is allowed to be above the target, that is, say that inverted gates are not allowed, or like the $q_{t} = (q_{c} + 1) \ mod \ N_q$ set in our experiment where $q_{t}$ and $q_{c}$ is the corresponding qubit position of the target and control operation, and $N_q$ denotes the number of qubits. Through these rules, we sacrifice the generated circuits' diversity to a small extent but ensure that the encoding's uniqueness is not destroyed, making our representation learning process effective. To display limitations more clearly in our paper, we have added a description of the limitations to the appendix of the revised version **(page 14)**.
>
> > **Q2:**
> > The notations around (2) are inconsistent. $MLP^{(s)}$ and $MLP(M^{(S)})$ seem to be the same thing.
>
> We are sorry for the mistake and we have modified this minor point in the revised version.
>
> > **Q3:**
> > The naming of “fidelity of quantum states” is not conventional. It’s better to use the name “state preparation”.
>
> Thank you very much for your correction. We also agree that state preparation is a more accurate term and have modified it in the revised version.
>
> > **Q4:**
> > The details in the training of the GINs learning of circuit representations are missing in the appendix.
>
> Thanks for your question, we have described the construction and key parameters of the GIN model in the paper. Of course, some hyperparameters are missing, so we have added more complete parameter information to the appendix in the revised version **(page 16)**.

---

### Author Response · Authors · 2023-11-21
**General Response**

We are very grateful to the reviewers for their constructive feedback. We have modified our paper based on feedback and have uploaded the revised version. We have marked the obvious modifications in the main body in blue. Except for Appendix A.2, all content in the appendix is newly added. We look forward to getting more feedback from reviewers.

---

### Author Response · Authors · 2023-11-22
**Gentle Reiminder**

Dear Reviewers,

We would like to express our gratitude for your valuable time and effort spent in providing detailed and constructive feedback on our paper. We have taken all your comments into consideration and have made necessary revisions to our paper as well as supplementary materials.

Please kindly note that we have responded to each of your queries and concerns separately. We have also updated our paper and supplementary materials in line with the recommendations given by you. We hope these updates have improved the clarity and completeness of our work.

The deadline of rebuttal is approaching (less than 24 hours), we kindly ask you to review our responses and the revised version of our work. We expect that these updates will address your concerns and improve the quality of our paper.

Again, we appreciate your invaluable feedback and look forward to your further comments and suggestions.

Best regards

Authors